# Combined BSA-Seq and RNA-Seq to Identify Potential Genes Regulating Fruit Size in Bottle Gourd (*Lagenaria siceraria* L.)

**DOI:** 10.3390/plants13152154

**Published:** 2024-08-03

**Authors:** Huarong Fang, Shishi Huang, Ruirui Li, Peng Wang, Qingwei Jiang, Chuan Zhong, Yanjuan Yang, Wenjin Yu

**Affiliations:** College of Agriculture, Guangxi University, Nanning 530004, China; 2017302001@st.gxu.edu.cn (H.F.); 2017391011@st.gxu.edu.cn (S.H.); 2231200530@st.gxu.edu.cn (R.L.); wangpeng@gxu.edu.cn (P.W.); 2017391013@st.gxu.edu.cn (Q.J.); zc@gxu.edu.cn (C.Z.); yjyang85@126.com (Y.Y.)

**Keywords:** bottle gourd, fruit size, BSA-seq, transcriptome

## Abstract

Fruit size is a crucial agronomic trait in bottle gourd, impacting both yield and utility. Despite its significance, the regulatory mechanism governing fruit size in bottle gourd remains largely unknown. In this study, we used bottle gourd (small-fruited H28 and large-fruited H17) parent plants to measure the width and length of fruits at various developmental stages, revealing a single ‘S’ growth curve for fruit expansion. Paraffin section observations indicated that both cell number and size significantly influence bottle gourd fruit size. Through bulked segregant analysis and combined genotype–phenotype analysis, the candidate interval regulating fruit size was pinpointed to 17,747,353 bp–18,185,825 bp on chromosome 9, encompassing 0.44 Mb and including 44 genes. Parental fruits in the rapid expansion stage were subjected to RNA-seq, highlighting that differentially expressed genes were mainly enriched in pathways related to cell wall biosynthesis, sugar metabolism, and hormone signaling. Transcriptome and resequencing analysis, combined with gene function annotation, identified six genes within the localized region as potential regulators of fruit size. This study not only maps the candidate interval of genes influencing fruit size in bottle gourd through forward genetics, but also offers new insights into the potential molecular mechanisms underlying this trait through transcriptome analysis.

## 1. Introduction

Bottle gourd (*Lagenaria siceraria* L.), an annual trailing herb, is a popular melon crop with economic importance. Bottle gourd is adapted to warm climates and widely cultivated in tropical-to-temperate regions worldwide [1]. In China, it is primarily grown south of the Changjiang River, particularly in Zhejiang and Guangxi, with an annual planting area of about 133.3 km^2^ [2]. Bottle gourd has diverse applications: young fruits are consumed as vegetables for their fever-reducing and detoxifying properties, while mature fruits are processed into containers and musical instruments, in addition to being used in handicrafts. Additionally, due to its robust root system and strong grafting compatibility and resistance, bottle gourd seedlings are commonly used as rootstocks for cucurbit crops like watermelon, melon, and cucumber [3,4]. As a horticultural crop, bottle gourd’s commercial value is largely derived from its fruit, with fruit size being a critical trait influencing yield, utility, and consumer preference. Therefore, fruit size is a vital focus in bottle gourd breeding and has the attention of breeders.

The development of fleshy fruit involves complex physiological and biochemical changes. Following fertilization, fruit growth initiates with cell division, progresses through concurrent cell division and expansion to establish fruit set, and concludes with cell expansion. Variations in fruit size are influenced by both the number and size of cells [5]. Numerous factors regulate fruit size, such as cell wall metabolism, the cell cycle, and plant hormone signaling pathways. Cell wall metabolism involves enzymes like glycosyltransferase [6], β-galactosidase [7], and cellulose synthase [8]. The cell cycle is controlled by cell-cycle proteins [9] and cyclin-dependent kinases (CDKs) [10], which influence cell number and fruit size. Plant hormones, including auxin, cytokinin (CK), and gibberellin (GA), interact to impact fruit development and size, with regulatory factors linked to hormone synthesis being crucial. The auxin early-response gene family contains auxin/indoleacetic acid (*Aux/IAA*), Gretchen Hagen3 (*GH3*), and small auxin up RNA (*SAUR*) [11]. *Aux/IAA* interacts with *TIR1/AFB*, a component of E3 ubiquitin ligase, allowing the release or inhibition of auxin response factors (*ARFs*). In tomato, silencing *SlAux/IAA17* increases cell size and results in larger fruits [12]. *ARF9* negatively regulates cell division during early fruit development and reduces fruit size [13]. *ARF106*, highly expressed during the cell division and expansion stages, positively regulates apple fruit size by modulating auxin signaling. *GH3* binds to amino acids to remove free auxin, and its low expression is associated with high auxin concentration during rapid fruit expansion [11]. *MdSAUR36* negatively regulates pericarp cell division, affecting apple fruit size [14]. *SAUR041* is a candidate regulator for grape fruit size, and is essential for cell expansion during fruit ripening [15]. In loquat, *EjSAUR22* responds to auxins and regulates cell size and fruit expansion [16]. Cytokinin, a primary hormone in plant growth and development, promotes the expansion and growth of kiwi fruit cells [17]. Cytokinin oxidase/dehydrogenase 5 (*ZjCKX5*) inhibits jujube fruit development, and its overexpression leads to smaller fruits. The transcription factors *ZjWRKY23* and *ZjWRKY40* target *ZjCKX5*, downregulating its expression and thereby increasing fruit size in red jujube [18]. The overexpression of *AtCKX2* in tomatoes reduces endogenous cytokinin levels in fruit tissue, resulting in thinner peel and smaller fruits [19]. *GRAS24*, which is involved in GA and auxin signaling, affects tomato fruit development by inhibiting cell division and expansion [20]. The peak period of GA content coincides with the cell division and expansion phases in fruit, highlighting GA’s crucial role in regulating fruit size [21]. Ethylene, a well-known inhibitor of certain tissue and cell growth, is synthesized by 1-aminocyclopropane-1-carboxylate synthase (ACS2). Mutants of *acs2* in cucumber and melon produce less ethylene, leading to reduced cell division and ultimately affecting fruit development and size [22,23]. *S1DREB3* negatively regulates abscisic acid (ABA) response, and its overexpression alters tomato fruit size [24].

Several transcription factors have been identified as key regulators of fruit size, including members of the ethylene response factor (*ERF*), *AP2*, *MADS-box*, *NAC*, *WOX*, *YABBY*, *bHLH*, and *MYB* families. The APETALA2/ethylene responsive factor (*AP2/ERF*) named *PavRAV2* directly represses the expression of *PavKLUH*, controlling sweet cherry fruit size [25]. In apple, the overexpression of microRNA172 inhibits *AP2* transcription, leading to cell shrinkage and a significant reduction in fruit size [26]. Similarly, silencing *MdMADS8* or *MdMADS9* significantly reduces cell and fruit size [27]. In tomato, the overexpression of the *SlNAC* transcription factor activated by AP3/PI 1 (*SlNAP1*) negatively regulates fruit size and weight [28]. Analysis of the *SlLAM1* gene, a member of the WOX family, shows that its loss of function can lead to changes in fruit size [29]. Mutation in *YABBY* causes its depressed expression and promotes polyventricle formation, thereby increasing fruit size [30]. *RSL4*, a *bHLH* transcription factor, transmits auxin signals to stimulate cell growth [31]. Three *R2R3-MYB* transcription factors—*SlFSB1*, *ZmMYB40*, and *ZmMYB95*—interact to regulate the differentiation and expansion of fruit cells [32]. In grape, *VvCEB1* is specifically expressed in fruit tissues during the expansion phase and influences the expression of genes related to cell expansion, including those involved in auxin metabolism and signal transduction [33]. In citrus, *CsMYB77* negatively regulates fruit ripening and size by modulating abscisic acid and auxin signaling pathways [34].

Fruit size is typically a quantitative trait measured by indicators such as length (L), width (W), L/W ratio, circumference, volume, and weight [35]. Functional genes regulating fruit size have been extensively studied in tomato, including *CNR/FW2.2*, *KLUH/FW3.3*, *SUN*, *OVATE*, *LOCULE NUMBER* (*LC*), *WUSCHEL* (*WUS*), *FASCIATED* (*FAS*), *CLAVATA* (*CLV*), *CRC*, *GRAS2*, and so on. *Fruit Weight 2.2* (*FW2.2*) and *Fruit Weight 3.2* (*FW3.2*) were among the first cloned genes involved in regulating cell division and cell number. *FW2.2*, a member of the cell number regulator (*CNR*) family, affects fruit growth via intercellular transport mechanisms [36,37]. *FW3.2* encodes the *KLUH* gene of the *CYP78A* subfamily of P450 enzymes, promoting an increase in cell number in fruit peel tissue and influencing fruit size [38]. *SUN* and *OVATE* are pivotal in fruit elongation because they affect cell division [39]. *LC* (coding for *SlWUS*) and *FAS* (coding for *SlCLV3*) loci are key factors in fruit size variation, regulating size by influencing the number of locules. *ENO*, a member of the *AP2/ERF* superfamily, works synergistically with mutations in *SlWUS* and *SlCLV3* to promote cell proliferation by regulating flower meristem activity, thus playing a significant role in increasing fruit size [40]. *CRABS CLAW* (*CRC*), belonging to *YABBY*, is crucial for carpel development; *SlCRCa* suppresses cell division by regulating related genes and inhibits cell expansion by regulating expansion protein genes and the GA pathway, thereby negatively regulating tomato fruit size [41]. The *FAS* locus inversions at *YABBY* and *SlCLV3* reduce gene expression and locule number [42]. Silencing *GRAS2* inhibits ovary growth and cell expansion, resulting in smaller fruits and lower weight [43]. In addition, cell size regulator (*CSR*), identified as a key determinant of tomato fruit weight, has been cloned and characterized, and the *CSR-D* allele increases fruit weight primarily by enlarging pericarp cell size rather than increasing the number of cell layers [44]. *SlPZF1*, encoding a zinc finger protein family member (*C2H2*), is preferentially expressed in the exocarp during tomato fruit development, and can control fruit size by influencing pericarp cell size and mediating the cell-cycle regulatory factors interacting with it [45]. 

Many advances have been made in identifying genes that regulate fruit size in melon crops. Weng et al. [46] identified 12 quantitative trait loci (QTLs) for fruit size (*FS1.1*, *FS1.2*, *FS2.1*, *FS2.2*, *FS3.1*, *FS3.2*, *FS3.3*, *FS4.1*, *FS5.1*, *FS6.1*, *FS6.2*, and *FS7.1*). In cucumber, the *CsSUN* at the *FS1.2* locus is a homolog of the tomato fruit-shape gene *SUN* [47]. *CsACS* is a candidate gene affecting slender cucumber fruit development [48]. *CsFUL1* negatively regulates fruit length by influencing the cell division and expansion mediated by *CsSUP* and regulating the expression of *CsPIN1* and *CsPIN7* [49]. *SF2*, encoding histone deacetylase, targets genes related to hormone synthesis and signal response pathways, and influences cell division and fruit length by regulating the balance of ethylene, cytokinin, and polyamine [50]. *Short-fruit 1* (*sf1*) is a single recessive gene that likely regulates fruit length through various hormone biosynthesis and signal transduction pathways [51]. In melon, genes such as *SUN*, *OVATE*, *LC*, *FAS*, *CNR/FW2.2*, and *SlKLUH/FW3.3* have also been identified to control fruit size [52]. Compared to other Cucurbitaceae crops, research on bottle gourd fruit shape has only emerged in recent years. Xu et al. [53] assembled a high-quality reference genome ‘ZAAS_Lsic_2.0′ for bottle gourd and located a dominant QTL controlling fruit shape on chromosome 6. Zhang et al. [54] conducted transcriptome analysis on two gourd species with diverse fruit sizes, revealing significant differences in the expression of genes related to cell wall metabolism, the cell cycle, and phytohormones, with *ERF* family transcription factors being the most abundant. However, to date, no specific genes related to fruit size in bottle gourd have been cloned. 

In this study, bulked segregant analysis (BSA) combined with transcriptome sequencing (RNA-seq) was utilized to identify key genes regulating bottle gourd fruit size. A small-fruited gourd (H28) was used as the female parent and a large-fruited gourd (H17) as the male parent to measure fruit size at different growth stages. This allowed for the analysis of dynamic changes in fruit size and differences in cell number and size between the parental pericarps through microscopic observation. Then, we initially localized an interval regulating gourd fruit size, expanded the F_2_ population, and developed single-nucleotide polymorphism (SNP) and Insertion–Deletion (InDel) markers within the candidate interval to refine the mapping range. Through BSA-based mapping and RNA-seq, combined with gene annotation, six genes were finally predicted as candidate genes. This is not only helpful for identifying genes regulating fruit size in bottle gourd, but also lays the foundation for elucidating the genetic mechanism underlying this trait. Furthermore, it provides valuable theoretical and practical insight for the molecular-assisted breeding of bottle gourd, accelerating the development of new cultivars with optimized fruit size.

## 2. Results

### 2.1. Phenotypic Evaluation of Fruit Size in Bottle Gourd

The fruit size of the parent plants in this study exhibited visible variations (Figure 1), which were analyzed by measuring the transverse diameter (maximum width of the fruit) and longitudinal diameter (length of the fruit from top to bottom) of the parent and F_1_ fruits. The results demonstrated significant differences in fruit dimensions between the female H28 and male H17 plants, with H28 producing smaller fruits than H17 (Table 1). Specifically, H28 fruits had a width of 48.47 ± 3.73 mm and a length of 67.99 ± 3.41 mm, categorizing them as small fruits. In contrast, H17 fruits had a width of 163.18 ± 9.63 mm and a length of 295.76 ± 49.23 mm, categorizing them as large fruits. The F_1_ fruits measured 121.32 ± 7.60 mm in width and 176.02 ± 32.46 mm in length, with mean values between the parents favoring the large-fruited H17.

### 2.2. Dynamic Changes in Fruit Development Process in Bottle Gourd

The fruit widths and lengths of H28, H17, and their F_1_ progeny were measured every three days from 3 days before pollination (−3 DAP) to 30 days after pollination (DAP). Measurements were taken over twelve periods, with ten fruits from different plants measured at each period to calculate the average values. These averages were used to plot the parental growth curves and analyze the dynamic changes in the growth processes of the parental fruits (Figure 2). As the fruits developed, both the fruit widths and lengths of H28, H17, and F_1_ gradually increased, with the differences between the parents becoming increasingly significant. Starting from −3 DAP, the fruit sizes of H28 and H17 exhibited noticeable differences, with H17 consistently having larger widths and lengths than H28. The size of F_1_ was between the two parents, but gradually tended towards H17. The most substantial growth in H28′s fruit width and length occurred at 0 DAP-9 DAP, during which the growth rate was at its peak. The average width of H28 fruits increased from 8.39 mm to 42.16 mm, while the average length grew from 16.62 mm to 62.10 mm. From 9 DAP to 15 DAP, the growth rate slowed down, and the fruit dimensions stabilized by 15 DAP. At maturity (30 DAP), the average fruit width and length of H28 were 48.04 mm and 67.54 mm, respectively. Meanwhile, the fruit size of H17 exhibited rapid growth from 0 DAP to 15 DAP, with the average width increasing from 12.00 mm to 146.00 mm and the average length from 30.59 mm to 230.99 mm. From 15 DAP to 24 DAP, the growth rate slowed, and the fruit width and length stabilized after 24 DAP. At maturity (30 DAP), the average width and length of H17 fruits were 168.37 mm and 280.89 mm, respectively. Consequently, the difference in fruit width and length between H28 and H17 reached its maximum at 30 DAP.

### 2.3. Paraffin Section Observation of Fruit Skin in Bottle Gourd

The growth change curves of the gourd fruits revealed significant differences in fruit width and length between H28 and H17. To further investigate these differences at the cellular level, bottle gourd flesh samples were collected at −3, 0, 3, 6, 12, 18, 24, and 30 DAP. These samples were prepared as paraffin slices in transverse and longitudinal sections to analyze the cell number and area at eight different developmental stages. As the gourd fruits developed and expanded, the cell area in both the transverse and longitudinal sections gradually increased (Figure 3A, Appendix A). In the transverse and longitudinal sections of the parental fruit skin, the cell number in the large-fruited H17 was significantly higher than in the small-fruited H28 from −3 DAP onward, while the cell area in H17 was significantly smaller than in H28 from 0 DAP (Figure 3B, Appendix A). These results indicate that both cell number and cell area are critical factors contributing to the differences in fruit size between H28 and H17, with cell number having an earlier impact on fruit size.

### 2.4. Mapping of Candidate Genes Related to Fruit Size in Bottle Gourd

Through BSA analysis, a total of 43.98 Gb of clean data was obtained from the parent and extreme pools, with each sample achieving a Q30 percentage above 91% and a GC content over 33%. The comparison of clean reads from the parent and F_2_ samples to the reference genome ‘ZAAS_Lsic_2.0′ showed an average alignment efficiency of 98.86%, an average coverage depth of 30×, and a genome coverage of 99.29% (with at least one base covered). These results confirm the sequencing data’s high quality and suitability for subsequent mutation detection and correlation analysis. A total of 771,950 SNPs were obtained from the four pools, including 294,636 high-quality SNPs used to calculate the SNP index between the two extreme pools. Preliminary BSA-seq mapping results, using the Euclidean Distance (ED) and ΔSNP-index algorithms, located genes regulating bottle gourd fruit size within the 13,950,000 bp to 19,960,000 bp interval on Chr9, spanning 6.01 Mb (Figure 4A). This region contained 642 genes, including 180 non-synonymous genes.

To further narrow the candidate range, four KASP markers were developed every 1–2 Mb within the initial region, and 500 F_2_ plants were genotyped. Using phenotypic analysis, the fruit size interval was refined to between markers KS16.5 (16,536,349 bp) and KS18.5 (18,589,323 bp), covering a length of 2.05 Mb. Subsequently, the F_2_ population was expanded to 3000 plants, and several InDel markers (with a base number difference ≥3 bp) were further developed to screen recombinant plants through genotypic–phenotypic analysis. Finally, the candidate interval for fruit size was positioned between markers IS17.74 (17,747,353 bp) and IS18.18 (18,185,825 bp), with five and two recombinant plants on the left and right sides, respectively (Figure 4B). The refined region spans 0.44 Mb and contains 44 genes, 10 of which exhibit non-synonymous mutations in both parents.

### 2.5. Comparison of H28 and H17 Fruit Transcriptomes

To determine the reason for the fruit size difference between H28 and H17, we collected the fruits at the period of rapid fruit expansion (6 DAP) and compared their transcriptomes. A total of 22.74 GB of clean data was obtained (an average of 5.68 Gb per sample), with each sample achieving a Q30 percentage over 93% (Table 2), indicating high-quality sequencing and suitability for further analysis. The clean reads from each sample were compared with the ‘ZAAS_Lsic_2.0’ reference genome, identifying 1,166 differentially expressed genes (DEGs). 

Gene expression analysis using DESeq identified DEGs between H28 and H17 based on two criteria: an absolute value of log2FoldChange >1 and a significance *p*-value < 0.05. The results revealed a total of 2433 DEGs, with 1632 genes up-regulated and 801 genes down-regulated in the small-fruited H28 (Figure 5A,B). To elucidate the main functions of these DEGs, Gene Ontology (GO) categories and Kyoto Encyclopedia of Genes and Genomes (KEGG) pathways were analyzed. The GO analysis showed that the DEGs were enriched in cellular components and biological processes, with a significant number involved in biological processes such as cell wall development and plant post-embryonic morphogenesis (Figure 5C, D). The KEGG analysis indicated that DEGs were enriched in metabolic pathways, including sugar metabolism, phytohormone signaling, photosynthesis, and MAPK signaling (Figure 5E).

### 2.6. Candidate Gene Prediction of Fruit Size

Based on the resequencing results, the sequences of 44 genes within the 0.44 Mb candidate interval were initially analyzed. Ten genes (*HG_GLEAN_10001518*, *HG_GLEAN_10001525*, *HG_GLEAN_10001529*, *HG_GLEAN_10001543*, *HG_GLEAN_10001546*, *HG_GLEAN_10001548*, *HG_GLEAN_10001549*, *HG_GLEAN_10001550*, *HG_GLEAN_10001556*, and *HG_GLEAN_10001558*) were found to have non-synonymous mutations in both parents. Through RNA-seq, the expression of all candidate genes within the mapped interval was analyzed, revealing that four genes (*HG_GLEAN_10001522*, *HG_GLEAN_10001544*, *HG_GLEAN_10001548*, and *HG_GLEAN_10001561*) were differentially expressed between the parents. In total, 13 genes had either non-synonymous mutations or differential expression (Appendix A), with *HG_GLEAN_10001548* exhibiting both. With the use of functional annotation, six genes were finally predicted as candidate regulators of fruit size in bottle gourd (Table 3).

## 3. Discussion

Fruit size in bottle gourd is an important quality trait and a key indicator of yield and economic value. Understanding the dynamic changes in fruit development at the physiological level is essential for studying fruit size differences in bottle gourd. In this study, the size of large-fruited H17 and small-fruited H28 was measured and analyzed at different developmental stages. The results indicated that the growth pattern of bottle gourd fruits follows a single ‘S’ curve: initially increasing and then flattening. Fruit size differences between the parents were evident at the ovary stage (−3 DAP). The fruit width and length continued to increase, with the differences peaking at 30 DAP, consistent with Yan et al.’s findings on fruit size variation in wax gourd [55]. In addition, many studies have shown that cell division and expansion determine the number and size of fruit cells, influencing overall fruit size. Studies in blueberries [56] and loquat [57] suggest that cell size affects final fruit size, while cell number is a more crucial factor in horticultural crops like apple [58], pear [59], sweet cherry [60], and plum [61]. The cytological observation of paraffin sections revealed that large-fruited H17 had significantly more cells than small-fruited H28 at the ovary stage (−3 DAP), consistent with the initial size differences observed. Significant differences in cell area between the parents were also observed from 0 DAP. Previous studies indicate that post-fertilization fruit growth starts with cell division, followed by continuous cell division and expansion to promote fruit formation. Our study confirms these findings. Importantly, it was observed that H17 fruit enlargement is influenced by both cell number and cell area, with cell number being the more critical factor. Therefore, cell division is likely the primary contributor to fruit size differences in bottle gourd, suggesting it is a key target for future breeding efforts.

In horticultural plant research, exploring the genes and molecular mechanisms regulating fruit size is a hot topic. However, little is known about the genes influencing bottle gourd fruit size. In this study, BSA-seq was employed to map the candidate genes within the range of 17,747,353 bp to 18,185,825 bp on Chr9, covering a physical distance of 0.44 Mb. This interval contained 44 genes, 10 of which had non-synonymous mutations. Subsequently, RNA-seq identified four DEGs within this interval. With the addition of functional annotation, six genes were predicted to regulate fruit size in bottle gourd. Among these, *HG_GLEAN_10001548*, which has both non-synonymous mutations and upregulated expression, encodes a protein with a pentapeptide repeat (PPR) sequence. PPR proteins are crucial for mitochondrial function and nuclear development, acting as essential RNA-binding proteins in plants and influencing the expression of organelle mRNA transcripts [62]. *ClaPPRs* have been implicated in regulating fruit development and ripening in watermelon [63]. The QTL *fw3.2* in tomato, which controls fruit weight, has been linked with seven putative genes, among which *ORF4* encodes a protein that is highly consistent with Arabidopsis *PNM1*, belonging to the PPR family [64,65]. Similarly, the *Ca12g10030* (PPR) site in pepper is associated with fruit weight, with the loss of PPR function leading to fruit development defects and seed sterility [66,67,68]. These findings emphasize the pivotal role of PPR proteins in controlling fruit size and weight. Additionally, *HG_GLEAN_10001518* encodes the transcription factor *DIVARICATA* subtype X1, belonging to the MYB family, that is known to regulate the uneven distribution of auxin signals and participate in ABA signal transduction [69]. *HG_GLEAN_10001525* encodes the cell division control protein 6 homology B-like protein, which is involved in DNA replication and controlled by cyclin-dependent kinase (CDK) phosphorylation [70]. *HG_GLEAN_10001544* is annotated as zinc transporter 6, which is a member of the ZIP protein family and is involved in the uptake, transport, and distribution of metal ions, influencing fruit growth and development [71,72,73,74]. *HG_GLEAN_10001556* and *HG_GLEAN_10001558* both encode UDP-glycosyltransferase 87A1-like (UGT) proteins, which are involved in hormone regulation and secondary metabolism, impacting cell wall metabolism and cell size increase. The overexpression of *GSA1*, which encodes UGT, can increase grain size in rice, while Arabidopsis *UGT75D1* negatively regulates epidermal cell growth in cotyledons [75,76]. Cytokinins primarily exist as glycosides in plants, and the overexpression of rice *UGT Os6* in Arabidopsis thaliana significantly increases cytokinin glycoside content. Although these predicted genes in the mapped interval are promising, further studies are required to pinpoint the key gene regulating fruit size in bottle gourd.

In this study, GO analysis revealed that DEGs are mainly enriched in processes such as cell wall development and plant postembryonic morphogenesis. KEGG analysis indicated enrichment in pathways related to sugar metabolism, phytohormone signaling, and MAPK signaling, aligning with previous research findings [77]. It has been shown that cell wall development significantly influences cell division and expansion during fruit development. Representative cell wall proteins and modifications can regulate cell size during plant growth and fruit development [78,79]. Since the cell wall is primarily composed of polysaccharides, the sugar metabolism pathway likely affects cell wall biosynthesis, thereby influencing cell division and expansion [80]. Hormones also play a critical role in regulating fruit size and development, often acting in concert. Several functional genes and transcriptional regulators, including *MYB* and *bZIP*, which are related to the candidate genes in this study, have been implicated in the hormonal regulation of fruit size. *MYB* is involved in auxin biosynthesis [81]. For instance, the exogenous application of GA3 to grape berries upregulated *vrax2* expression (a member of the *R2R3-MYB* family), and the overexpression of *vrax2* increased fruit size [82]. Similarly, *bZIP* controls cell elongation by regulating the biosynthesis and transduction of auxin and GA [83,84]. Therefore, future research on bottle gourd fruit size should focus on the key pathways related to cell wall synthesis and metabolism, as well as plant hormones, to identify genes or transcription factors controlling fruit size and explore the underlying regulatory mechanisms.

## 4. Conclusions

We used BSA-seq combined with RNA-seq to identify key genes regulating the fruit size of bottle gourd. The fine mapping of the dominant locus revealed that it was on Chr9 in a 0.44 Mb interval containing 44 genes. Six genes within the localized region were predicted to be candidate regulators associated with bottle gourd fruit size through the analysis of sequence variation and transcriptional differences, together with gene annotation. Our findings will aid the cloning of genes controlling bottle gourd fruit size, and provide information about the potential molecular mechanisms of the candidate genes regulating fruit size from a transcriptomic perspective.

## 5. Materials and Methods

### 5.1. Plant Materials and Phenotypic Evaluation of Fruit Size

Bottle gourd self-inbred lines H28 and H17, preserved by the Vegetable Research Group at the Agricultural College of Guangxi University, were chosen as the female (P_1_) and male (P_2_) parents, respectively. H28 produces small fruits with an average width of 48.47 mm and a length of 67.99 mm, whereas H17 produces large fruits with an average width of 163.18 mm and a length of 295.76 mm. Both parent lines exhibit a cucurbit shape, but with significant differences in fruit size (Figure 1). H28 and H17 were crossed to obtain the F_1_ generation, which was subsequently self-crossed to conduct an F_2_ segregating population. All plants were cultivated in the experimental field of Guangxi University in 2021. 

Fruits at full maturity (40 DAP) were selected, and their transverse diameter (maximum width of the fruit) and longitudinal diameter (length of the fruit from top to bottom) were measured using an electronic vernier caliper with an accuracy of 0.01 mm. These measurements were used to evaluate the mature fruit size of H28 and H17, with three fruits measured per plant to calculate the average value. Additionally, parental fruits were collected at twelve developmental stages: −3, 0, 3, 6, 9, 12, 15, 18, 21, 24, 27, and 30 days after pollination (DAP). The width and length of the parent fruits were measured at each stage to compare dynamic size changes during fruit development, with three fruits measured to obtain an average in each period. 

### 5.2. Paraffin Section and Cytological Observation

The parent fruits were harvested at eight representative developmental stages (−3, 0, 3, 6, 12, 18, 24, and 30 DAP). The lower and middle parts of the bottle gourd fruits were selected, the skin was scraped off, and the flesh tissue was cut into approximately 2 cm^3^ cubes for transverse and longitudinal paraffin sectioning. The paraffin sectioning process involved fixing and dehydrating the pre-treated samples, embedding them in paraffin, and sectioning them using a microtome (RM2016 Leica, Wetzlar, Germany). The prepared sections were then dewaxed and stained. Observations were made using a light microscope (Nikon Eclipse E100, Tokyo, Japan), and panoramic scanning software (Slide Viewer 2.5) was used to select appropriate views and take pictures. Three views per period were randomly selected to count the cell numbers, and thirty cells per view with complete and clear boundaries were selected to calculate the cell area. The data were collated and analyzed using ImageJ 2023, Excel 2021, and SPSS 18.

### 5.3. BSA-Seq Mapping Approach

Thirty plants with the largest fruits and thirty plants with the smallest fruits were selected from five hundred F_2_ plants. Young leaves were collected from these sixty plants to construct extreme mixed pools. Association analysis was conducted on the two F_2_ pools and the parental pools using the ‘ZAAS_Lsic_2.0’ genome as the reference. Pooled DNA samples were prepared for library construction and sequenced on the Illumina HiSeq™PE150 platform (San Diego, CA, USA). The raw reads obtained from high-throughput sequencing were analyzed and converted to sequencing reads after base calling. These reads were then filtered to obtain clean reads, ensuring quality for subsequent analysis. The clean reads were compared with the reference genome for mutation detection. Finally, the Euclidean Distance (ED) and ΔSNP-index methods identified the regions associated with the target traits [85,86].

### 5.4. Fine Mapping 

To narrow the preliminary region, Kompetitive Allele-Specific PCR (KASP) markers were designed for each 1–2 Mb distance within the candidate interval based on BSA-seq data. PCR amplification was prepared following the manufacturer’s instructions (LGC Genomics, Shanghai, China). The PCR reaction volume was 3 μL, consisting of 1.0 μL of DNA (8–15 ng μL^−1^), 1.5 μL of 2× master mix, and 0.5 μL of primer mix. The amplification used landing PCR with the following conditions: heat treatment at 95 °C for 15 min; denaturation at 95 °C for 20 s; annealing and extension between 65 and 55 °C for 1 min, 10 cycles, reducing 1.0 °C each cycle; denaturation at 95 °C for 20 s; annealing and extension at 57 °C for 1 min, 26 cycles; then holding at 4 °C in the dark. After amplification, fluorescence scanning and genotyping were performed. Four pairs of KASP markers were designed to identify recombinant plants from the 500 F_2_ plants. Subsequently, the population was expanded to 3000 plants, and new InDel markers within the mapped interval were developed to screen recombinant plants and determine their genotypes. Finally, the most likely candidate region was inferred using genotype–phenotype analysis. Primers are listed in Appendix A.

### 5.5. RNA Sequencing (RNA-Seq) Analysis

Fruit flesh from both parents at the rapid fruit expansion stage (6 DAP) was selected and sent to Paisonol Company (Shanghai, China) for RNA-seq and bioinformatics analysis. Total RNA was extracted from the tissue samples, and its purity, quantity, and integrity were assessed using a NanoDrop and Agilent 2100 bioanalyzer (Thermo Fisher Scientific, Waltham, MA, USA). Library construction and high-throughput sequencing were performed on the Illumina HiSeq2500 platform (San Diego, CA, USA). Clean data were filtered from raw data using SOAPnuke v1.5.6 and compared with the reference genome ‘ZAAS_Lsic_2.0’ using HISAT2 v2.0.4 and Bowtie2 v2.2.5. DESeq2 software was used to analyze the differentially expressed genes (DEGs) related to fruit size and enrich the GO [87] and KEGG [88] pathways in the parents.

## Figures and Tables

**Figure 1 plants-13-02154-f001:**
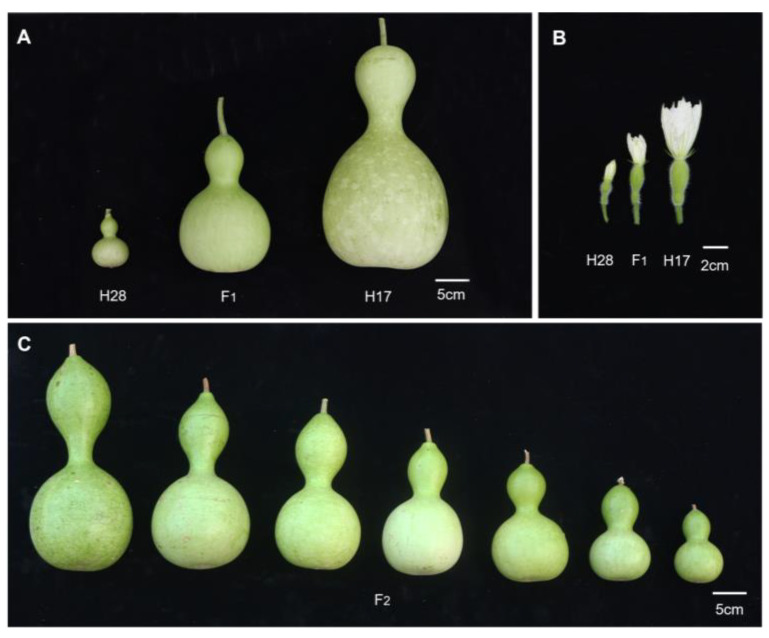
Bottle gourd fruits: male H28, female H17, their F_1_, and some F_2_. (**A**) Mature fruit of parent and F_1_. (**B**) Ovary of parent and F_1_. (**C**) Mature fruit of some F_2_.

**Figure 2 plants-13-02154-f002:**
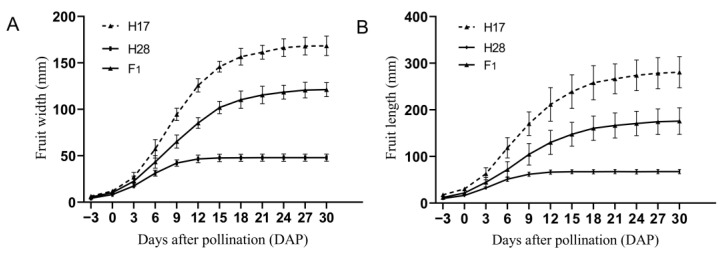
Dynamic changes in the fruit growth process in bottle gourd. (**A**) The growth curve of fruit width of parent and their F_1_ from −3 DAP to 30 DAP. (**B**) The growth curve of fruit length of parent and their F_1_ from −3 DAP to 30 DAP.

**Figure 3 plants-13-02154-f003:**
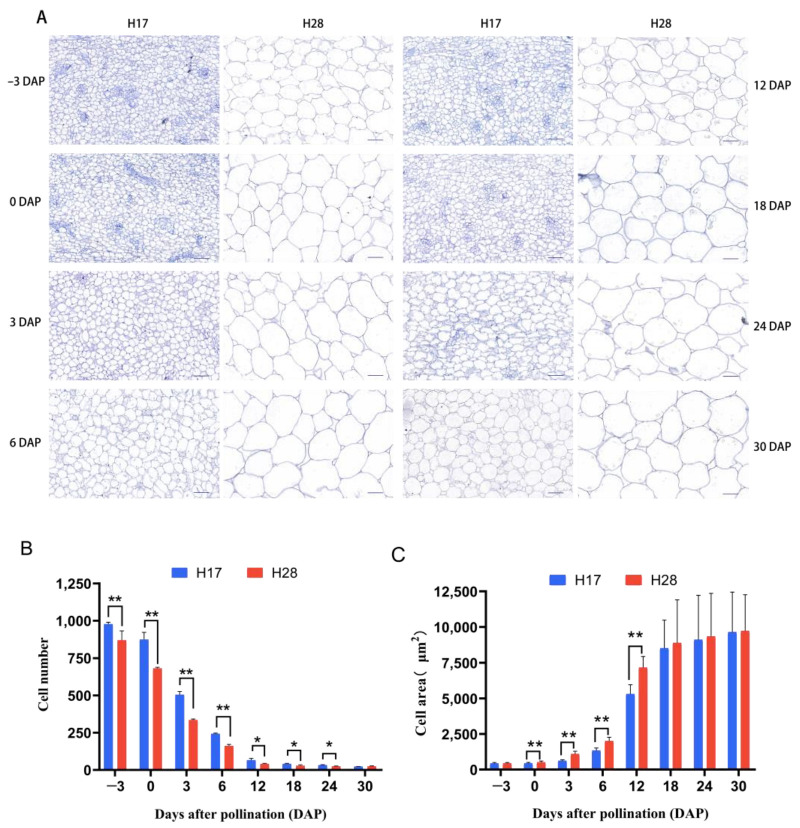
Microscopic observation and comparison of the average cell area and number in transverse sections of H17 and H28 fruit at different periods. (**A**) Paraffin section of H17 and H28 fruit from −3 DAP to 30 DAP; bar = 50 μm. (**B**) Statistical diagram of cell number. (**C**) Statistical diagram of cell area. *, *p* < 0.05; **, *p* < 0.01.

**Figure 4 plants-13-02154-f004:**
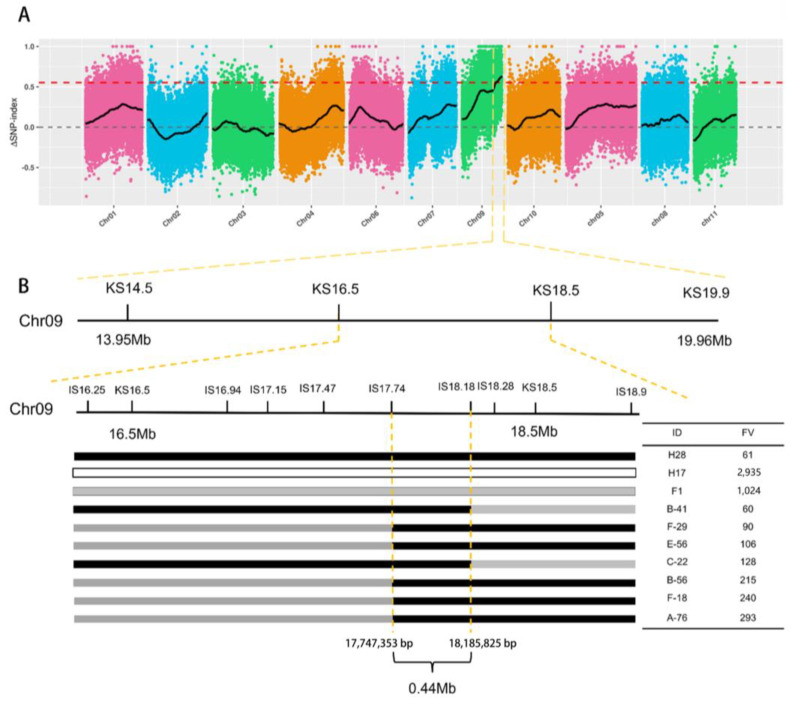
Mapping of fruit size in bottle gourd. (**A**) Chromosome distribution in terms of ΔSNP-index correlation value. The colored dots represent the calculated ΔSNP-index values, the black lines represent the fitted ΔSNP-index values, and the red line represents the 99% threshold line. (**B**) Candidate interval of fruit-size genes in bottle gourd. Black represents the genotype of small-fruit H28, white represents the genotype of large-fruit H17, and gray represents the F_1_ genotype.

**Figure 5 plants-13-02154-f005:**
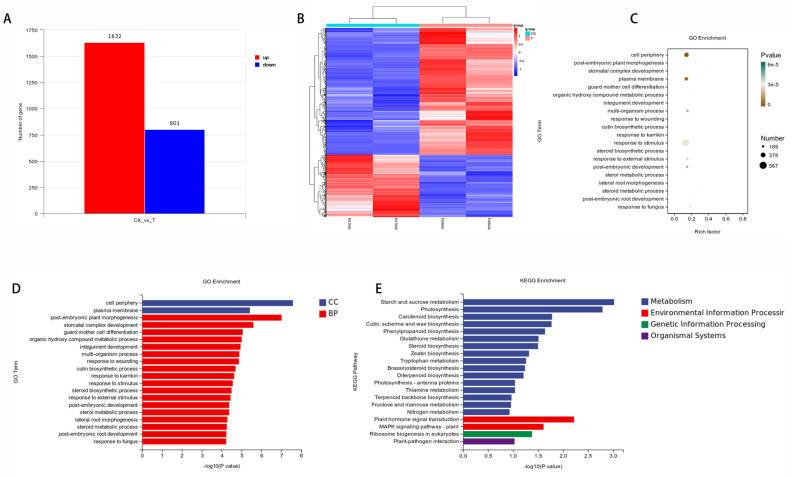
Transcriptomic analysis of fruit size of H28 and H17. (**A**) Number of DEGs. (**B**) Heat map showing the expression pattern of genes in blue and red tissues. Horizontal represents genes and each column represents a sample, with red representing highly expressed genes and green representing low-expressed genes. (**C**) Rich factor of GO enrichment analysis of DEGs. (**D**) Histogram of GO enrichment analysis of DEGs. (**E**) Histogram of KEGG enrichment analysis of DEGs.

**Table 1 plants-13-02154-t001:** Analysis of width and length of bottle gourd fruit in parents and F_1_.

Population	Fruit Width (mm)	Fruit Length (mm)
P_1_ (H28)	48.47 ± 3.73	67.99 ± 3.41
P_2_ (H17)	163.18 ± 9.63	295.76 ± 49.23
F_1_ (H28 × H17)	121.32 ± 7.60	176.02 ± 32.46

**Table 2 plants-13-02154-t002:** RNA-Seq reads obtained from H28 and H17 fruits.

Sample	Clean Reads (Strip)	Clean Bases (bp)	Q30 (%)
H286d1	39,275,150	5,930,547,650	94.13
H286d2	38,048,230	5,745,282,730	94.46
H176d1	39,540,490	5,970,613,990	93.97
H176d2	33,727,920	5,092,915,920	94.05

**Table 3 plants-13-02154-t003:** Functional annotation and relative expression mode of candidate genes.

Gene ID	Gene Function Annotation	Non-Synonymous Mutation	Relative Expression Mode
HG_GLEAN_10001518	Transcription factor *DIVARICATA* subtype X1, *MYB* family	YES	Nodiff
HG_GLEAN_10001525	Cell division control protein 6 homology B-like protein	YES	Nodiff
HG_GLEAN_10001544	Zinc transporter 6, ZIP protein family	NO	Upregulation
HG_GLEAN_10001548	Pentapeptide-repeat (PPR) sequence protein	YES	Upregulation
HG_GLEAN_10001556	UDP-glycosyltransferase 87A1-like (UGT) protein	YES	Nodiff
HG_GLEAN_10001558	UDP-glycosyltransferase 87A1-like (UGT) protein	YES	Nodiff

## Data Availability

The data presented in this study are available in this article and in the Appendix A.

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
