# Peer review of "Combined BSA-Seq and RNA-Seq to Identify Potential Genes Regulating Fruit Size in Bottle Gourd (Lagenaria siceraria L.)"

_plants, 2024, doi:10.3390/plants13152154_

Round 1

Reviewer 1 Report

Comments and Suggestions for Authors

In the present study, the authors have utilized a cross-based method for rapid trait mapping (BSA) combined with RNA-seq to identify key genes regulating the fruit size of bottle gourd.  Results predicted six genes as candidate genes regulating fruit size.

Comments:

1. The authors should include the data for F1/F2 fruits (Dynamic changes in the fruit growth process in bottle gourd) along with their parent in Fig 2 as these data were presented in Fig 1 and Table 1, as well as Fig 6 and Table 2 (along with Fig 5).

2. Fig 3: could be represented as I marked directly in the manuscript. The data for longitudinal sections of H17 and H28 fruit should be presented here. Or this data was presented in Supplementary Figure 1. I do not know whether this data was presented since I could not access the Supplementary files.

3. Section “4.4. Fine mapping”. Please double-check the PCR running conditions. I do not see the extension temperature of PCR. And why did the authors hold in the dark “then holding at 4 °C in the dark” (Line 429)?

4. Lack of citation information on methods used for evaluating data in some sections of Material and Methods

For example: in the section “4.3 BSA-seq mapping approach”., the Euclidean Distance (ED) and ΔSNP-index methods lack citation. Provide the references.

5. The authors should provide more detailed information of six genes (that were predicted to regulate fruit size in bottle gourd) and present in the context of the manuscript or at least put the data in the Supplementary. The detailed information including gene name, etc. so that the audience can assess this information, instead of presenting only gene ID such as HG_GLEAN_10001518.

6. This manuscript lacks the section “Conclusion”. Since this section is not mandatory, but it is better If the authors include it.

Minor remarks:

- Lowercase “Single-Nucleotide-Polymorphism” (Line159)

- Capitalize Each Word for the article title and all sections as the journal format.

For example: “2.1. Phenotypic evaluation of fruit size in bottle gourd” -> “2.1. Phenotypic Evaluation of Fruit Size in Bottle Gourd” 

- Change “GB” to “Gb” (Line 232, 263).

- Use a uniform presentation for mathematics data: for example “771,950” or “771.950”

Author Response

Comments 1:  The authors should include the data for F1/F2 fruits (Dynamic changes in the fruit growth process in bottle gourd) along with their parent in Fig 2 as these data were presented in Fig 1 and Table 1, as well as Fig 6 and Table 2 (along with Fig 5).

Response 1: Thank you for pointing this out. The data (Dynamic changes in the fruit growth process in bottle gourd) for the parents has been put in the manuscript, and the F1 data was not previously put in, but now it has been added in Figure 2. In addition, growth curves of fruits were produced to clarify the process of fruit development. The materials of P1, P2, and F1 respectively represents dominant pure, recessive pure, and heterozygous genotypes, and the F2 population contains all the genotypes in P1, P2, and F1. Therefore, using P1, P2, and F1 fruits to produce growth curves has been able to illustrate the effect of the primary effector loci on bottle gourd fruit size in this study.

Remark:The revised sections in the whole text are marked in red font in the re-uploaded manuscript (Please see the attachment.)

Comments 2:  Fig 3: could be represented as I marked directly in the manuscript. The data for longitudinal sections of H17 and H28 fruit should be presented here. Or this data was presented in Supplementary Figure 1. I do not know whether this data was presented since I could not access the Supplementary files.

Response 2: Thank you very much for your valuable suggestions. Figure 3 has been revised as you directly marked. The data for longitudinal sections of H17 and H28 fruits have been uploaded as Supplementary Figure S1 in the original Supplementary files.

Comments 3:  Section “4.4. Fine mapping”. Please double-check the PCR running conditions. I do not see the extension temperature of PCR. And why did the authors hold in the dark “then holding at 4 °C in the dark” (Line 429)?

Response 3: The extension temperature of the PCR is in Lines 427-428 of the text, which reads “annealing and extension between 65 and 55 °C for 1 min, 10 cycles, reducing 1. 0 °C”. In addition, “then holding at 4 °C in the dark” is to indicate the storage conditions for subsequent genotyping.

Comments 4:  Lack of citation information on methods used for evaluating data in some sections of Material and Methods. For example: in the section “4.3 BSA-seq mapping approach”, the Euclidean Distance (ED) and ΔSNP-index methods lack citation. Provide the references.

Response 4: Thank you for your suggestion. The citation information on methods used to evaluate the data have been added to the Materials and Methods in Lines 422 and 448-489.

Comments 5:  The authors should provide more detailed information of six genes (that were predicted to regulate fruit size in bottle gourd) and present in the context of the manuscript or at least put the data in the Supplementary. The detailed information including gene name, etc. so that the audience can assess this information, instead of presenting only gene ID such as HG_GLEAN_10001518.

Response 5: Thank you for pointing this out. More detailed information about the six genes (including gene name, physical location, mutation information, gene annotation from various databases, etc.) has been uploaded to the Supplementary file before.

Comments 6:  This manuscript lacks the section “Conclusion”. Since this section is not mandatory, but it is better If the authors include it.

Response 6: Thanks for your suggestion. We have added “Conclusion” to the manuscript in Lines 369-377.

Comments 7:  Minor remarks:

- Lowercase “Single-Nucleotide-Polymorphism” (Line159)

- Capitalize Each Word for the article title and all sections as the journal format.

For example: “2.1. Phenotypic evaluation of fruit size in bottle gourd” -> “2.1. Phenotypic Evaluation of Fruit Size in Bottle Gourd” 

Change “GB” to “Gb” (Line 232, 263).

Response 7: Thank you for pointing this out. We have modified these mistakes as you remarked.

Reviewer 2 Report

Comments and Suggestions for Authors

The reviewed manuscript, titled "Combined BSA-seq and RNA-seq to identify potential genes regulating fruit size in bottle gourd (Lagenaria siceraria L.)” by Fang et al. is a work that is primarily on identifying and characterizing the genetic underpinning that result in the size differences seen between two strains of the agriculturally important crop that is known as the bottle gourd.  Overall, the authors designed and presented an interesting study that pinpoints the differences seen between a small and large strain to a single chromosomal region, and they characterize the transcriptional differences via RNA sequencing.  This work is interesting, and it is one that I believe adds to the literature and would be of interest in the field to a number of researchers.  That said, there are a few revisions that this reviewer believes are necessary prior to publication of this work.

The introduction is quite detailed in a way that results in the authors losing focus and conciseness.  The section would benefit from streamlining and consolidating the many specific genes discussed in a number of specific difference plant species into a more focused format that summarizes the relevant parts that the author are trying to make.

Line 181: table 1 – include the n values for the sampling populations that you utilized to generate the average.  Specify what the +/- represents (SD, variance, etc.).

Line 205: figure legend is incomplete, should be completely informative as a standalone section allowing the reader to completely interpret and understand the work.  As written, this is incomplete.  Further, the two strains that are presented quantitatively seem like they have been characterized previously and thus this figure seems like a control, at best.  Since the authors have crossed these to generate an F1 generation of hybrids, the fact that those plants were not included in this seems confusing to me.  That strikes me as a design flaw and omission that required remediation.

Line 221: This figure is poorly presented.  Add the time point to each of the appropriate panels.  Add relevant details to allow us to interpret (are all taken at the same magnification?  Where are the scale bars?).  Rather than presenting this as a qualitative figure – the authors should measure a representative population of the cells to quantify the size differences seen in each strain at each stage.  This can easily be done and then the cell size can be averaged, the SD/SE calculated, and the differences presented as a graph that will be much more rigorous than a simple series of selective pictures.  This appears to be what is done in figure 3 – making these as two separate figures unnecessary.  These two figures should be merged into a single one.  Additionally, the F1 hybrids are missing from this – as this on its own has limited novel addition to the literature, the lack of the hybrid measurements is surprising and seems quite relevant to be added.

Line 256: Revise the figure legend to be complete.

In the Discussion section the authors have presented some informative information that shows that there is a very specific chromosomal region that appears to be responsible for the size differences seen in each of the mature fruits.  This reviewer has a very strong interest in the genomic organization – and I have authored a number of papers that are focused on the functional genomic clustering of genes.  I am including one such reference below – and as it is one that I am an author on, there is no pressure to have these authors include this work in their manuscript, however there are many appropriate others in the literature that the authors can use.  I would like the authors to describe the functional relationships and significance of the genes that dictate size being colocalized in this manner and address this as thoroughly as possible.

Reference:

Cittadino, G.M.; Andrews, J.; Purewal, H.; Estanislao Acuña Avila, P.; Arnone, J.T. Functional Clustering of Metabolically Related Genes Is Conserved across DikaryaJ. Fungi 20239, 523. https://doi.org/10.3390/jof9050523

Comments on the Quality of English Language

English language needs minor revisions only.

Author Response

Comments 1:  The introduction is quite detailed in a way that results in the authors losing focus and conciseness.  The section would benefit from streamlining and consolidating the many specific genes discussed in a number of specific difference plant species into a more focused format that summarizes the relevant parts that the author are trying to make.

Response 1: Thank you for pointing this out. The introduction has been appropriately revised to streamline and consolidate the genes discussed for the same species. And the revised sections in the whole text are marked in red font in the re-uploaded manuscript (Please see the attachment.).

Comments 2:  Line 181: table 1 – include the n values for the sampling populations that you utilized to generate the average.  Specify what the +/- represents (SD, variance, etc.).

Response 2: Thank you for your question. The +/- after the average stands for standard deviation (SD).

Comments 3:  Line 205: figure legend is incomplete, should be completely informative as a standalone section allowing the reader to completely interpret and understand the work.  As written, this is incomplete.  Further, the two strains that are presented quantitatively seem like they have been characterized previously and thus this figure seems like a control, at best.  Since the authors have crossed these to generate an F1 generation of hybrids, the fact that those plants were not included in this seems confusing to me.  That strikes me as a design flaw and omission that required remediation.

Response 3: Thank you very much for your valuable suggestions. The legend in Figure 2 has been modified. The F1 data on dynamic changes in the fruit growth process was not previously put in the result, but now it has been added in Figure 2.

Comments 4:  Line 221: This figure is poorly presented.  Add the time point to each of the appropriate panels.  Add relevant details to allow us to interpret (are all taken at the same magnification?  Where are the scale bars?).  Rather than presenting this as a qualitative figure – the authors should measure a representative population of the cells to quantify the size differences seen in each strain at each stage.  This can easily be done and then the cell size can be averaged, the SD/SE calculated, and the differences presented as a graph that will be much more rigorous than a simple series of selective pictures.  This appears to be what is done in figure 3 – making these as two separate figures unnecessary.  These two figures should be merged into a single one.  Additionally, the F1 hybrids are missing from this – as this on its own has limited novel addition to the literature, the lack of the hybrid measurements is surprising and seems quite relevant to be added.

Response 4: 

Thank you for pointing this out. Figure 3 has been modified by adding the time points to each appropriate panel. Each figure in Figure 3 is at the same magnification, and the bar is added to the bottom right corner of each figure.

In addition, we merged Figure 3 and Figure 4 into single one, and quantified the size differences in the fruit at each stage by counting the number and area of cells in each paraffin section and presented the differences as a statistical diagram in new Figure 3.

In this paper, the purpose of observing fruit cells through paraffin sections was to clarify the regulatory effects of key genes regulating fruit size at the cellular level. Therefore, paraffin sections of homozygous alleles(P1 and P2)could well achieve the experimental objectives, so we just chose P1 and P2 fruits to make paraffin sections for observation.

Comments 5:  Line 256: Revise the figure legend to be complete.

Response 5: Thank you for your advice. The legend of Figure 4 has been modified to be complete.

Comments 6:  In the Discussion section the authors have presented some informative information that shows that there is a very specific chromosomal region that appears to be responsible for the size differences seen in each of the mature fruits.  This reviewer has a very strong interest in the genomic organization – and I have authored a number of papers that are focused on the functional genomic clustering of genes.  I am including one such reference below – and as it is one that I am an author on, there is no pressure to have these authors include this work in their manuscript, however there are many appropriate others in the literature that the authors can use.  I would like the authors to describe the functional relationships and significance of the genes that dictate size being colocalized in this manner and address this as thoroughly as possible.

Response 6: Thank you very much for your advice. As mentioned in your reference, the functional genomic clustering of genes is very important for studying the gene relationships and molecular mechanisms regulating fruit size in bottle gourd. In this study, we mainly report a mapping region that regulates the fruit size in bottle gourd and predict six candidate genes. For gene function analysis, we will continue to map clone the candidate gene and characterize them by transgenic experiments in subsequent study, so as to elucidate the regulatory mechanism of fruit size in bottle gourd.

Round 2

Reviewer 1 Report

Comments and Suggestions for Authors

Dear the authors,

Thank you for your revising.

The comments raised by me have been addressed.

Except, you might forget to change “GB” to “Gb” (Line 254, revised version).